# The Nexus between Fire and Soil Bacterial Diversity in the African Miombo Woodlands of Niassa Special Reserve, Mozambique

**DOI:** 10.3390/microorganisms9081562

**Published:** 2021-07-22

**Authors:** Ivete Sandra Alberto Maquia, Paula Fareleira, Isabel Videira e. Castro, Ricardo Soares, Denise R. A. Brito, Aires Afonso Mbanze, Aniceto Chaúque, Cristina Máguas, Obinna T. Ezeokoli, Natasha Sofia Ribeiro, Isabel Marques, Ana I. Ribeiro-Barros

**Affiliations:** 1Forest Research Center, School of Agriculture, University of Lisbon, Tapada da Ajuda, 1349-017 Lisbon, Portugal; ivete.s.maquia@uem.ac.mz; 2TropiKMan Doctoral Program, NOVA SBE, 2775-405 Carcavelos, Portugal; 3Biotechnology Center, Eduardo Mondlane University, Maputo 3453, Mozambique; denise.brito@uem.mz; 4National Institute of Agricultural and Veterinary Research, I.P. (INIAV, I.P), 2780-157 Oeiras, Portugal; paula.fareleira@iniav.pt (P.F.); isabel.castro@iniav.pt (I.V.e.C.); ricardo.soares@iniav.pt (R.S.); 5Faculty of Agrarian Science, Lúrio University, Niassa 3302, Mozambique; airesmbanze@unilurio.ac.mz; 6Faculty of Agronomy and Forest Engineering, Eduardo Mondlane University, Maputo 3453, Mozambique; achauque2012@gmail.com (A.C.); joluci2000@yahoo.com (N.S.R.); 7Centre for Ecology, Evolution and Environmental Changes (cE3c), Faculty of Sciences, University of Lisbon, 1749-016 Lisbon, Portugal; cmhanson@fc.ul.pt; 8Department of Microbiology and Biochemistry, University of the Free State, Bloemfontein 9300, South Africa; 2019875893@ufs4life.ac.za

**Keywords:** Miombo, fire, 16SrRNA, rhizosphere, plant growth promoting bacteria, *Brachystegia boehmii*

## Abstract

(1) Background: the Miombo woodlands comprise the most important vegetation from southern Africa and are dominated by tree legumes with an ecology highly driven by fires. Here, we report on the characterization of bacterial communities from the rhizosphere of *Brachystegia boehmii* in different soil types from areas subjected to different regimes. (2) Methods: bacterial communities were identified through Illumina MiSeq sequencing (16S rRNA). *Vigna unguiculata* was used as a trap to capture nitrogen-fixing bacteria and culture-dependent methods in selective media were used to isolate plant growth promoting bacteria (PGPB). PGP traits were analysed and molecular taxonomy of the purified isolates was performed. (3) Results: Bacterial communities in the Miombo rhizosphere are highly diverse and driven by soil type and fire regime. Independent of the soil or fire regime, the functional diversity was high, and the different consortia maintained the general functions. A diverse pool of diazotrophs was isolated, and included symbiotic (e.g., *Mesorhizobium* sp., *Neorhizobium galegae*, *Rhizobium* sp., and *Ensifer* adhaerens), and non-symbiotic (e.g., *Agrobacterium* sp., *Burkholderia* sp., *Cohnella* sp., *Microvirga* sp., *Pseudomonas* sp., and *Stenotrophomonas* sp.) bacteria. Several isolates presented cumulative PGP traits. (4) Conclusions: Although the dynamics of bacterial communities from the Miombo rhizosphere is driven by fire, the maintenance of high levels of diversity and functions remain unchanged, constituting a source of promising bacteria in terms of plant-beneficial activities such as mobilization and acquisition of nutrients, mitigation of abiotic stress, and modulation of plant hormone levels.

## 1. Introduction

Soil microorganisms play crucial roles in ecosystem functioning, contributing significantly to soil structure and fertility as well as to plant responses to environmental drivers [1,2,3]. In association with climate and anthropogenic factors, the soil microbiome is a major ecological determinant of forest ecosystems [4]. One gram of forest soil harbors millions to billions of bacteria that mediate the overall soil dynamics, indicating ecosystem health and land use [5,6,7,8].

Tropical soil ecosystems are particularly diverse and complex, having a unique set of microbial functions with recognized potential for developing bio-based solutions, and yet remain largely understudied [9,10]. The most relevant tropical ecosystems (i.e., Amazonia, Congo, New Guinea forests, and Miombo-Mopane woodlands) are ranked among the top High Biodiversity Wilderness Areas (HBWA) [11,12]. Miombo is the most important and widespread type of vegetation from southern Africa, covering approximately 2 million km^2^, i.e., 70% of the Sudan-Zambezian phytoregion [13]. Together with the Mopane eco-region (ca. 0.5 km^2^), Miombo is among the five hotspots of unique species endemism [11]. Dominated by legume trees belonging to the genera *Brachystegia*, *Julbernardia*, and *Isoberlinia* [14], Miombo is a major provider of ecosystem goods and services through the provision of timber and nontimber products to more than 150 million rural and urban dwellers in the poorest countries of the world (revised by [15]). However, the increasingly growing human population, together with unsustainable management practices and climate pressure, impose major modifications in the ecoregion at the expense of biodiversity and ecosystem services [16,17,18,19].

Soil formation and conservation are among the major regulation and supporting services in the Miombo woodlands [15]. Miombo soils are mostly nutrient-poor and dry [14]; thus, the accumulation and decomposition of organic matter are of utmost importance for maintaining soil structure and fertility [20,21]. Ultimately, wildfires resulting from the combined action of climate, human, and animal pressures are the major woodland’s drivers, undermining biodiversity composition, structure, and function [22,23], and may become major contributors to greenhouse gas (GHG) emissions [24]. At the same time, fires are important management tools [25] that are essential for the maintenance of the phytosociological structure of Miombo, as they contribute to regrowth and seed germination [15,26] as well as to soil mineralization and nutrient availability [27,28,29]. Thus, investigating fire and biodiversity dynamics in the Miombo woodlands is relevant to understand ecological processes. Considering that soil diversity is a key determinant of an ecosystem’s stability [1,2,3], and that the Miombo woodlands are dominated by tree legumes with high resiliency to extreme environmental conditions [14,15], the analysis of plant growth promoting bacteria (PGPB) is a crucial step to bridge the research gaps. PGPB comprise endosphere and rhizosphere bacteria that stimulate plant growth and development, and enhance plant performance [30]. These bacteria have been classified into four functional categories: (i) biofertilizers, which increase soil fertility and the accessibility of nutrients to plants; (ii) phytostimulators, which produce phytohormones that govern plant growth and development; (iii) bioremediators, which remove organic pollutants; and (iv) biocides, which mitigate the effect of pests and diseases [31].

Considering the global socio-ecological and economic importance of the Miombo woodlands and the inexistence of scientific studies addressing the complex nexus of factors driving the soil dynamics, in this paper we analyzed the effects of fire regime (high vs. low return intervals) and soil type (sandy, red, and oxic soils) in the diversity and functions of rhizosphere bacteria from the dominant Miombo legume tree, *Brachystegia boehmii* Taub. Three complementary approaches were used: (i) high throughput Illumina MiSeq sequencing of bacterial 16S rRNA gene; (ii) isolation of PGPB; and (iii) analysis of the effects of fire regime and soil type on PGP functions.

## 2. Materials and Methods

### 2.1. Site Description

The fieldwork was conducted in the Niassa Special Reserve (NSR; North Mozambique, Niassa Province) located between latitudes 12°38′048.67″ S and 11°27′05.83″ S and meridians 36°25′21.16″ E and 38°30′23.74″ E. NSR is the largest (approx. 42.000 km^2^) and the most pristine area of Miombo (WWF, 2014; Figure 1). The climate is tropical sub-humid, with an average rainfall of 900 mm that increases from the east (800 mm) to the west (1200 mm), and a mean annual temperature of 25 °C during the dry season (May–October) and 30 °C during the wet season (November–April) [32]. Miombo woodlands cover more than 70% of the total area of the NSR and are composed of approximately 8500 plant species, half of which are endemic. *Brachystegia* spp., *Julbernardia globiflora* Benth. (Troupin), *Dyplorhynchus condilocarpon* (Müll.Arg.) Pichon, and *Pseudolachnostylis maprouneifolia* Pax, dominate the canopy cover [33,34], and a dense and continuous grass layer dominates the forest floor. NSR also has a high diversity of faunal species, including elephants, pala-pala, lions, wild dogs, leopards, buffalo, and more than 400 bird species [35].

### 2.2. Sampling

The rhizosphere soil of *Brachystegia boehmii* was collected in three different soil types: brownish-gray sandy soils (sandy soils—S); red soils of medium texture (red soils—R); and red oxi-soils of medium texture (oxic soils—O) (Table 1). Considering each type of soils, samples were collected in sites with two different fire frequency regimes: low fire frequency (L, fire return interval > 7 years) and high frequency (H, fire return interval < 1 year) (Figure 1). SH and SL depict samples from brownish-gray sandy soils for high and low fire frequency regime, respectively, RH and RL represent samples from red soils of medium texture for high and low frequency regimes, respectively, while OH and OL indicate samples from red oxi-soils of medium texture for high and low fire frequency regimes, respectively. Soil samples were collected at approximately 20 cm depth, in eight circular plots of 40 m diameter arranged in two contiguous parallel transects using the Riley and Barber’s shake method [36]. A total of 24 rhizosphere soil samples were collected (4 replicates × 3 soil types × 2 fire frequencies) and preserved at −80 °C until DNA extraction.

### 2.3. DNA Extraction, Amplification, and Sequencing of 16S rRNA Genes

DNA was extracted from 100 mg soil samples using the DNeasy PowerSoil Kit (Qiagen, Germantown, MD, USA), following the manufacturer’s instructions. Library preparation and Illumina MiSeq paired-end 300 bp sequencing run was performed by Macrogen, Seoul, South Korea. Briefly, DNA was first quantified by PicoGreen (Invitrogen, Waltham, MA, USA) using Victor 3 fluorometry, and the quality was determined on a 2100 Bioanalyzer (Agilent technologies, Santa Clara, CA, USA). To analyze the bacterial composition of the samples, the V3–V4 region of the 16S rRNA gene was amplified by polymerase chain reaction (PCR) under the following conditions: initial denaturation at 98 °C for 2 min, followed by 35 cycles of denaturation at 95 °C for 30 s, primer annealing at 53 °C for 40 s, extension at 72 °C for 1 min, and a final extension phase at 72 °C for 5 min. Specific primers 341F (5′-CCT ACG GGG NGG CWG CAG-3′) and 805R (5′-GAC TAC HVG GGT ATC TAA TCC-3′) and their adapters at the 5′ and 3′ ends of the DNA fragments were used [37]. The sizes and amounts of PCR-enriched fragments were verified on a 2100 Bioanalyzer (Agilent Technologies, Santa Clara, CA, USA). The libraries were normalized before sequencing with the Illumina MiSeq system (Illumina, San Diego, CA, USA).

### 2.4. Assembly of Reads and Taxonomical Assignment

Raw sequences were checked and trimmed using Quantitative Insight Into Microbial Ecology (QIIME) v.1.7.0 [38] to remove short and low-quality sequences that contained less than 200 bp, ambiguous bases, and had a minimum quality score of 20. Clean paired-end reads were merged using FLASH v. 1.2.11 [39]. Potential chimeric sequences were identified and removed by the UCHIME algorithm [40]. Unique sequences were grouped in operational taxonomic units (OTUs) with 97% similarity using VSEARCH v.6.1.544 [37]. Pruning of OTUs with a low number of sequences (<5) was conducted on a per-sample basis, as an OTU that is common in one sample may occur as a low-abundance contaminant in others due to cross-contamination. The most abundant sequence of each OTU was selected as representative. Taxonomy was assigned by searching for similar sequences, conducted with BLAST v2.2.29 using a similarity coefficient of 98%, against the Greengenes 13_5 online database [41].

### 2.5. Diversity Analysis

QIIME v.1.9.1 platform was used to calculate the alpha diversity index of Shannon-Wiener (H’) with the package alpha_diversity. The software was also used to produce rarefaction curves to compare the richness of different OTUs using the package alfa_rarefaction.py. Beta diversity was estimated by computing a phylogenetic tree based on unweighted UniFrac distances [42]. The UniFrac distance matrix was visualized using principal coordinates analysis (PCoA). A PERMANOVA test (999 permutations) based on a Bray–Curtis distance matrix calculated on normalized OTU counts was conducted to determine the effect of soil and fire frequency on bacterial community structure. A Kruskal–Wallis or a Mann–Whitney test was performed using Statistica v13.3 (StatSoft, Tulsa, OK, USA) [43] to determine the effects of soil type and fire frequency, respectively, on the OTU richness and diversity. Linear discriminant analysis (LDA) effect size (LEfSe) was performed to identify taxa with differential abundances (LDA score > 2.0; Kruskal-Wallis *p* < 0.05) between samples [44]. Thereafter, the top 200 differential features were summarized as a taxonomic cladogram using GraPhlAn v.1.1.3.1 [45].

### 2.6. Functional Prediction

Functional predictions of the bacterial community were conducted by AllGenetics & Biology SL (www.allgenetics.eu) using the software PICRUSt2 [46]. We used the hidden state prediction tool, implemented in the castor R package [47] to normalize the data and predict gene families’ profiles. To generate a finer resolution of the predicted changes within the metagenomes, we weighted the results of the prediction of gene families by the relative abundance of the OTUs to infer the metagenomes of the community, using the OTU abundances per sample. We mapped enzyme commission (EC) abundances onto gene pathways using MinPath [48] and the MetaCyc reactions database [49]. The resulting pathway abundance table represents how much each OTU contributed to the community-wide pathway abundance. We used the package STAMP [50] to perform statistical analysis of the functional profiles recovered using the two-sided Welch’s *t*-test [51]. We compared multiple groups of samples (red vs. sandy vs. oxic soil) by computing an ANOVA/Tukey-Kramer test to determine significant differences between the metabolic pathways detected in each group. The Benjamini-Hochberg FDR [52] correction method was applied to the results and all features with an adjusted *p*-value ≥ 0.05 were removed. To easily visualize the spread of the data in each group, we generated a principal component analysis (PCA) plot.

### 2.7. Isolation and Characterization of Root Nodule Bacteria Using a Trap Host Legume

The promiscuous tropical legume plant, *Vigna unguiculata* (cowpea), obtained from a local landrace from Manica, Mozambique (GUR; see [53,54]), was used as a trap host for rhizobia bacteria, following the protocol described by [9]. Briefly, seeds were surface-sterilized, soaked in sterilized water, and germinated in an agar and water solution of 0.8% (weight/volume). Seedlings were transferred to jars containing inert sand and 50 mL of N-free Jensen plant nutrient medium [55]. A diluted soil solution (1:10) from each sample was independently applied on the roots of individual seedlings (1 mL/seedling). A non-inoculated positive N-control supplied with 1.75% KNO_3_ (TN) and a non-inoculated N-free negative control supplemented with ¼ liquid Jensen medium control (T0) were also prepared. Five replicates were performed for each soil type. Plants were grown in a growth chamber (photoperiod of 16 h day and 8 h night and temperature 23 °C day and 18 °C night) for 4 weeks and in a greenhouse for a further 7 weeks. Plants were watered with liquid Jensen medium once per week. The presence of nodules and the nodulation phenotype were examined after 11 weeks. The dry weights of shoots were calculated and used to determine the index of symbiotic effectiveness (Es), according to Ferreira and Marques [56]: Es = (Xs − XT0/XTN − XT0) × 100, where Xs represents the mean dry weight of inoculated plants ± SD; XTN represents the mean dry weight of plants from nitrogen control and XT0 represents the mean dry weight of uninoculated plants. The data were analyzed by one-way analysis of variance (ANOVA) with the Statistica v. 13.0 software (StatSoft, Tulsa, OK, USA), using the Fisher’s least significant difference (LSD) test at *p* ≤ 0.05.

Root nodules were then surface-sterilized and crushed. A droplet of the nodule suspension was streaked on a yeast mannitol agar (YMA) plate containing Congo red [57]. Plates were incubated at 28 °C in the dark and bacterial growth was followed for 10 days. Isolate purity was checked by examining the colony morphology and Congo red absorption. Subculturing was conducted when more than one type of colony was present.

### 2.8. Isolation of Rhizosphere Bacteria and Characterization of In Vitro Plant Growth Promoting Activities

All rhizosphere samples (SH, SL, OH, OL, RH, and RL) were used for the isolation of soil bacteria according to [9]. Briefly, 2 g aliquots of the soil samples were suspended in 18 mL of 0.85% NaCl, serially diluted (ten-fold), and plated on Tryptone-Yeast Agar (TYA) [58] and Burk’s agar [59]. Plates were incubated at 30 °C and single colonies were selected based on their morphology and repeatedly transferred to fresh agar plates until purified. Isolated bacteria were routinely grown in Tryptone-Yeast (TY) broth and TYA. 

Soil bacterial isolates were evaluated for in vitro activities related to the promotion of plant growth using the same procedures as described previously [60]. For evaluation of aerobic growth in the absence of nitrogen, the isolates were plated on Burk’s N-free agar and growth was evaluated after incubation for 5–7 days at 30 °C. The production of auxins was evaluated according to [61], as described in [60]. Siderophore production was determined by the chrome-azurol (CAS) assay [62], modified by [63]. The formation of orange haloes around colonies was considered indicative of siderophore production [62]. The ability to solubilize mineral phosphate was evaluated in yeast-extract-dextrose (YED) agar plates supplemented with 5 g L^−1^ Ca_3_(PO_4_)_2_ [64]. Cellulase activity was assayed on agar plates as described by [63], according to the methods of [65,66,67]. Briefly, bacteria were inoculated on plates containing N-Fixing bacteria (NFb) agar medium supplemented with 0.5% tryptone and 0.2% carboxymethylcellulose. After incubation at 30 °C for three days, the plates were covered with Congo red solution (1 mg mL^−1^) for 30 min to facilitate the distinction of orange halos around the bacterial growth.

## 3. Results and Discussion

### 3.1. Sequencing and Distribution of OTUs

A total of 5,429,179 raw 16S rRNA gene sequence reads were obtained from all 24 samples. After removal of low-quality sequences and chimeras, 629,400 sequence reads were obtained with an average of 26,225 sequences per sample and 57.83% GC content (average of four replicates; Appendix A). Rarefaction curves for each of the samples reached a plateau, indicating sufficient depth to capture 16S rRNA gene diversity (Appendix A).

Red soils from high fire frequency sites (RH) presented the highest number of OTUs (1391) and oxi-soils from low fire frequency sites (OL) presented the lowest number (977: Figure 2A). Nevertheless, independent of the soil type and fire regime, the observed changes were not statistically significant (H = 3.04, *p* = 0.22). However, the influence of fire on soil bacterial diversity was reflected by an increase in the Shannon index in all high-frequency fire sites (Figure 2; H = 1.81, *p* = 0.41). The PCA analysis showed some spatial separation of oxi-soils from the other samples (Figure 2B).

Altogether, these results have shown that the bacterial diversity in the Miombo rhizosphere is highly diverse, and apparently more conserved within oxi-soils (O). These contrast with a similar study performed in another important savanna ecosystem, the Mopane woodlands [9], where the rhizobacteria were found to be less abundant (700–800 OTUs vs ca. 1000–1200) and less diverse (Shannon H index ca. 4 vs ca. 7–8) than the results found here [9]. The results also contrasted with those observed in the subtropical Laurisilva forest, where fire drastically reduced bacterial abundance (from ca. 700–800 to ca. 300 OTUs) [68]. Thus, soil dynamics depend on a complex nexus of several factors, including anthropogenic and environmental parameters [27,69,70] that are probably site-specific. In the cases of the savanna ecosystems (including Miombo and Mopane), wildfires are among the major evolutionary drivers undermining the development of fire tolerance and recovery strategies in the endemic vegetation [71]. Our focal species, *Brachystegia boehmii*, is adapted to high fire frequencies, and there seems to be a pyrodiversity effect, i.e., an increase in genetic diversity driven by fire [72,73], which, according to our Shannon diversity results, is also associated with the rhizobacteria community [74,75], likely with a positive impact in fire tolerance.

### 3.2. Taxonomic Composition of the Microbial Communities

In total, OTUs represented 17 phyla, 50 classes, 104 orders, 212 families, 565 genera, and 1057 species. The most abundant phylum was *Actinobacteria* (39.49 ± 12.89%), followed by *Proteobacteria* (19.54 ± 7.25%) and *Acidobacteria* (12.75 ± 4.79%) that, together, accounted for more than 72.43 ± 5.05% of the relative abundance of sequence reads from all samples (Figure 2). An average of 7.31% OTUs could not be identified (S: 5.99%; R: 5.73%; O: 10.21%; Figure 2).

The abundance of *Actinobacteria* showed significant differences between soils, fire, and the interaction of both factors (Appendix A). *Actinobacteria* was negatively affected by fire in sandy (S) and red (R) soils, which was significantly more abundant under regimes of low fire frequency (LFF) than under high fire frequency (HFF) (Figure 3). In the case of oxi-soils (O), HFF promoted a higher abundance of *Actinobacteria*. The abundance of *Proteobacteria* was significantly affected by fire (Appendix A), which was more frequent under HFFs. The abundance of *Acidobacteria* showed no significant differences between fire frequencies, soils, and their interaction (Appendix A).

These phyla are frequently reported in the soils of other forest ecosystems [9,68,76,77] comprising ubiquitous bacteria with a multiplicity of ecological functions such as soil fertilization, bioremediation and plant nutrition, growth, and protection [1,78,79]. In general, the bacterial variation patterns imposed by HFF followed the same trend in S and R soils, contrasting with O soils. This was unexpected, as from a physical and chemical point of view, R and O soils are more similar, although all Miombo soils are generally nutrient-poor, acidic, and have a low cation exchange capacity [14]. Plausible explanations for this discrepancy may be the severity of the burnings as well as the phytosociological status at the sampling sites [15,68].

Among the classifiable phylotypes, three dominant bacterial genera were found across all soil samples, including *Gaiella* (8.82 ± 3.81%), *Chthoniobacter* (5.75 ± 5.29%), and *Solirubrobacter* (5.53 ± 2.98%), all which were significantly affected by fire, type of soil, and their interaction (Appendix A). HFF significantly decreased the abundance of *Gaiella*, except in O soils (Figure 4). Fire had no impact on the abundance of *Chthoniobacter* in S and R soils, but in O soils the abundance of this genus decreased significantly under HFF (Figure 4). *Solirubrobacter* was negatively affected by HFF in S and R soils but positively affected in O soils. The common major genus *Saccharopolyspora* was significantly affected by soil type (Appendix A) and was more abundant in S and R soils under LFF but negligible under HFF (Figure 4). The presence of this genus in O soils was minimal, with a slightly higher abundance under HFF compared to LFF regime. The abundance of the genus *Bacillus* was also affected by the type of soil (Appendix A) and was more abundant in S soils but significantly lower in R soils and negligible in O soils (Figure 4; Appendix A).

The ecological role of *Gaiella* is largely unknown, but studies suggest a close phylogenetic relationship with thermophilic, halotolerant, and radiotolerants bacteria [80,81]. Thus, its presence upon fire may be related to thermotolerance [82]. Additionally, the presence of this genus is widely reported and related to: (i) antibiotic resistance and denitrification, thus being a potential source of nitrous oxide, a major greenhouse gas [83]; (ii) production of phtalides, organic compounds with antimicrobial, insecticidal, cytotoxic, and anti-inflammatory properties, among others [84]; and (iii) metabolization of organic compounds and nutrient cycling [85,86]. Similarly, the few studies available regarding the environmental role of *Chthoniobacter* suggest its involvement in the transformation of organic compounds [87,88], possibly contributing to the cycling of carbon by the degradation of various complex carbohydrates, such as cellulose [89]. The same suggestion applies to the genus *Solirubrobacter* which has been described as the most dominant in different soil ecosystems, with functional analyses supporting a role in the nitrogen cycle and as a source of bioactive compounds [84]. The environmental role of *Brevitalea* is not yet described, but its occurrence was reported in savanna soils [90]. Recent studies suggest that the presence of *Rhodoplanes* in a consortia of rhizosphere bacteria is related to soil restauration [91,92,93,94] and nitrogen cycling [95,96], while the halophytic *Saccharopolyspora* that has been reported in several saline environments (salt pans, mountains, lakes, salterns, and in the date palm rhizosphere) is related to the production of biocompounds [97,98].

A total of 27 discriminative features (linear discriminant analysis (LDA) score > 2 and Kruskal-Wallis *p* < 0.05) were identified between soil samples following a sample-wide LDA effect size (LEfSe) analysis (data not shown). The top discriminative features are shown in Figure 5. Of these, the classifiable genus-level features differentially more abundant in OL were: *Acidibacter*, highly tolerant to transition metals, as well as to arsenic [99]; *Acidobacterium*, related to organic matter transformation in forest soils [100], which is considered an indicator for soil fertility and plant health [101]; *Desulfohalophilus*, described as extreme halophilic sulfate- and arsenate-respiring bacteria [102]; *Edaphobacter*, whose presence has been reported in heavy metal contaminated soils [103] and more recently indicated as a potential source for bioelectricity production from sludge matrices [104]; and *Paraburkholderia*, identified as the main nitrogen-fixing microsymbionts of *Mimosa tenuiflora* from Caatinga (tropical dry forests) [105]. *Actinokineospora* was differentially more abundant in OH. This genus constitutes a promising source of a wide range of bioactive compounds of pharmaceutical interest [106,107,108,109]. *Zavarzinella* is likely involved in the degradation of organic matter [110] and *Microlunatus,* a strong candidate to manage environmental pollution [111], was differentially abundant in RL, while *Pirellula*, a potential indicator of desertification [112], and *Virgisporangium*, of yet unknown function, were more abundant in RH. Finally, *Xylanimicrobium*, a novel petroleum hydrocarbon degrading bacterium [113], was differentially more abundant in SL, while there was no genus-level phylotype differentially abundant in SH compared with the other sites (Figure 5). These results reveal a clear impact of soil and fire on the plant’s rhizosphere, reinforcing the importance of the microbial flora of Miombo for new bio-based solutions.

PICRUSt2 identified 413 functional categories with a high degree of homology between samples (Appendix A). A top pathway (PWY-3781), which is involved in aerobic respiration I (cytochrome c), was found in all samples (Appendix A). However, this pathway was more abundant in soils from LFF sites than from HFF (F = 1.25; *p* = 0.027). The type of soil also influenced the abundance of this pathway (F = 1.33; *p* =0.022), which was more often under O, then followed by R, and to a lower extent, S soils (Appendix A). Soil respiration plays an important role in nutrient cycles, such as carbon and nitrogen, and is therefore an important indicator of environmental changes [114]. The results suggest an elevated rate of bacterial activity in the soils [115], reflecting the resilience of bacteria to poor soils and high temperatures [9]. Nevertheless, their contribution to GHG emissions should also be addressed with caution, as this is a major issue in the Miombo woodlands [24].

### 3.3. Vigna unguiculata as a Trap for Rhizobia Bacteria

Considering the limitations of NGS on species identification, we further used *Vigna unguiculata* (cowpea) as a trap host to identify PGP bacteria present in the soil. Inoculation of cowpea plants with soil samples increased shoot dry weight to levels comparable and even higher than those of control plants supplied with nitrogen (Figure 6; Appendix A), suggesting that the symbiosis with rhizobia bacteria was active.

Most Es values were higher than 75%, indicating the presence of rhizobia strains highly effective in nitrogen fixation in almost all cases, except for the rhizobial population of SH, which showed the lowest value (Es = 42.18%).

A total of 64 isolates were purified from *V*. *unguiculata* nodules, 82.8% of which belonged to the phylum *Proteobacteria* (53 isolates) and only 4.7% belonging to phylum *Firmicutes* (three isolates). The remaining 12.5% belonged to unidentified bacteria (eight unknown) (Appendix A). Fourteen genera were successfully identified, nine of which were nitrogen-fixers (diazotrophs), namely, the symbiotic *Mesorhizobium* sp. (six isolated: SL9B, RH5A, RH9A, RH13A, RH11A, and OH11A), *Neorhizobium galegae* (RL16A), *Rhizobium* sp. (seven isolates: SL3, RL10B, RL10C, RL11A, OL2A, OL6A, and OH3BC), *Ensifer adhaerens* (three isolates: RL17, RH14A, and RH18A); and the nonsymbiotic *Agrobacterium* sp. (10 isolates: SL9C, RL4C, RL5B, RL14B, RL20, RH6A, RH12A, OL5A, OH10A, and OH12A), *Cohnella* sp. (SL4), *Herbaspirillum huttiense* (SH3A), *Pseudomonas* sp. (four isolates: SH5A, RL11B, OL1A, and OL4), and *Stenotrophomonas* sp. (13 isolates: SL10AB, SL10AC, RL6A, OL2B, OL2C, OL7A, OL10A, OL12A, OL13A, OL14A, OL15A, OL16A, and OH7A). Most of these bacteria have important applications as PGP. These results confirm those obtained by other authors [116,117,118,119], who showed that in addition to rhizobia, other bacteria can frequently inhabit the interior of nodules without causing damage or disease and are thus considered nodule endophytic bacteria. In general, fire had a negative effect on the diversity of the trapped bacteria and bacterial richness was dependent on the soil type, with higher levels in R, intermediate levels in O, and lower levels in S. 

The *Stenotrophomonas* sp., particularly *S. maltophilia* was the predominant bacteria in the isolates in all soil types (SL10AC, RL6A, OL2B, OL2C, OL10A, OL12A, OL14A, OL15A, and OH7A), which was more prevalent in low frequency fire of oxic soil types and absent in sandy and red soils. Studies have demonstrated the potential of these bacteria in biological control against *Ralstonia solanacearum*, a bacterium that causes potato brown rot [120]. This ability as a biocontrol agent is related to antimicrobial and insecticidal properties due to the production of several bioactive compounds [121,122]. These bacteria secrete siderophores, improving iron availability [122], and are also important for bioremediation [122,123]. Additionally, they produce phytohormones that promote root and seedling growth [124,125].

The second most predominant bacteria were *Agrobacterium* sp. In addition to fixing nitrogen, this genus is also involved in the production of phytohormones [126] and species such as *A*. *tumefaciens* (OL5A) and *A*. *rhizogenes* (RL20) may be pathogenic, constituting important plant transformation tools [127,128,129].

*Rhizobium* sp. was the third most predominant genus found. Together with *Allorhizobium*, *Azorhizobium*, *Bradyrhizobium*, *Mesorhizobium*, and *Sinorhizobium* rhizobia, these bacteria form nitrogen-fixing root nodules on leguminous species, although in species belonging to Caesalpinioideae, as *B*. *boehmii*, this association seems to be rare [130,131] though it is poorly studied. Our results showed that the rhizosphere of *B*. *boehmii* has a high diversity regarding the rhizobia genera, which contrasts with other tree legumes, such as *Faidherbia albida*, *Albizia versicolor* [132] and *Colophospermum mopane* [9], where only *Bradyrhyzobium* has been identified.

Although less represented, *Pseudomonas* has also been isolated from trapping cowpea nodules and is indicated as a key PGPB [60]. Studies conducted on *Pelargonium graveolens* demonstrated the production of siderophore, phytohormones, and inorganic phosphate solubilization by *P*. *monteilii* (RL11B) [133] and growth promotion in *Arabidopsis thaliana* and *Lactuca sativa* by *P*. *nitritireducens* (OL4) [134].

*Variovorax* (RL5A, RL13A, RL14A, and RL19B) has been identified as PGPB in Chinese cabbage and green pepper through siderophores and 1-aminocyclopropane-1-carboxylic acid (ACC) deaminase production [135]. ACC deaminase additionally helps plant development under adverse environmental conditions [136].

Other species isolated were *Lactobacillus reuteri* (RH5B and RH8A), *Lysobacter soli* (RL8DB), *Rubrimonas shengliensis* (RL7A), and *Xanthomonas oryzae* pv. *oryzae* (SH1 and SH8). There are no studies describing the role of *L*. *reuteri*, *L*. *soli*, or *R*. *shengliensis* in soils; however, *X*. *oryzae* pv. *oryzae* is described as the most important rice pathogen [137]. This opens the door to new areas of research to understand the role of these bacteria in Miombo soils.

### 3.4. Characterization of In Vitro Plant Growth Promoting Activities and Taxonomic Analysis of Bacteria Isolated from Soils

Eight bacterial isolates with plant growth promoting activities were recovered from soils, including one isolate from sandy soil, three from red soils, and four from oxic soils (Table 2). Isolates from sandy and oxic soils were obtained with TYA as an isolation medium, whereas isolates from red soil were obtained using Burk’s N-free medium. Based on the sequence analysis of the 16S rRNA gene, all isolates were assigned to the phylum Proteobacteria, namely *Burkholderia* sp. (5RHB and 6RHB), *Caballeronia zhejiangensis* (4RHB), and *Variovorax defluvii* (10OHA), which belong to *Burkhoderiales*; and *Microvirga* sp. (10SLA) and *Rhizobium* sp. (4OLA, 5OLA, and 5OHA), which belongs to *Rhizobiales*.

All isolates were able to grow aerobically in a nitrogen-free medium (Table 2) and are therefore likely to be nitrogen fixers potentially contributing to the nitrogen uptake of *B*. *boehmii*. Although there are no studies that confirm nitrogen fixation by *C*. *zhejiangensis*, all other bacteria were previously described as nitrogen fixers [138,139,140,141,142,143]. The ability to solubilize phosphate was not detected in any of the isolates. Isolates from all soil types were able to produce the plant hormone indole-3-acetic acid (IAA) (Table 2), namely *Microvirga* sp. (10SLA) in sandy soils, *Burkholderia* sp. (6RHB) in red soils, and *Rhizobium* sp. (4OLA, 5OLA, and 5OHA) in oxic soils. The ability to produce phytohormones was demonstrated in *Microvirga ossetica* when isolated from root nodules of *Vicia alpestris* [144]. The capacity of *Burkholderia* sp. and *Rhizobium* sp. to produce IAA was also well described in other studies [126]. Production of siderophores was only detected in isolates from red soils, namely *Burkholderia* sp. (5RHB) and *C*. *zhejiangensis* (4RHB). Siderophore producing bacteria may act as bioremediators and thus help plants to tolerate heavy metal stress (Al, Cd, Cu, Fe, Ga, In, Pb, and Zn) [29,145,146,147,148,149]. It was previously demonstrated that a heavy metal-resistant *Burkholderia* sp. isolate, which was able to produce siderophores and perform several other plant growth promoting activities, could also increase the uptake of Pb and Cd in tomato and maize plants growing in contaminated soil [150]. Additionally, siderophores may also act in biocontrol since they can reduce the proliferation of phytopathogens by preventing the acquisition of iron [145,151]. Only *V*. *defluvii* (10OHA) isolated from oxic soils showed cellulase activity (Table 2). *Variovarax* sp. is described as a useful source of hydrolytic enzymes [152]. This suggests the involvement of this species in the control of plant pathogens [153].

## 4. Conclusions

The combination of next generation sequencing and culture dependent approaches allowed a comprehensive analysis of the bacterial diversity and associated functions in the rhizosphere of *Brachystegia boehmii*. The produced knowledge constitutes a significant contribution to the understanding of the dynamics of soil bacteria in tropical forest ecosystems, namely in plant-microbe interactions, and bioprospecting of new bacterial species. Although tropical ecosystems host ca. 80% of the global biodiversity, pressures imposed by a combination of human, animal, and climate pressure are inflicting major challenges. Thus, understanding key aspects of ecosystem dynamics is important to support community-based conservation programs.

The changes observed in the bacterial communities associated with the rhizosphere of *B*. *boehmii* are a consequence of a combined effect of soil type and fire regime. Nevertheless, independent of either factor, bacterial diversity was high, suggesting resiliency to extreme environments. The functional profiles of different bacterial consortia were consensual, safeguarding their main general functions. The most abundant taxonomic groups constituted novel sources of important traits, such as thermo- and halo-tolerance, production of biocompounds, and nutrient cycling. Several of these species may also represent new plant growth promoting bacteria (PGPB).

Similarly, the diversity of trapped bacteria in *Vigna unguiculata* was also dependent on soil type and fire frequency, and was considerably high, particularly when compared to other tropical legume trees. In all cases, the presence of diazotrophs (symbiotic: *Mesorhizobium* sp., *Neorhizobium galegae*, *Rhizobium* sp., and *Ensifer adhaerens*; and non-symbiotic: *Agrobacterium* sp., *Cohnella* sp., *Herbaspirillum huttiense*, *Pseudomonas* sp., and *Stenotrophomonas* sp.) was evident. Consistent with these results, all soil isolates were also able to grow in a nitrogen-free medium and thus are possibly nitrogen-fixers, presenting cumulative PGP functions, such as IAA production (*Microvirga* sp., *Burkholderia* sp., and *Rhizobium* sp.), siderophore production (*Burkholderia* sp. and *C*. *zhejiangensis*), and plant protection abilities (*Variovarax defluvii*).

In conclusion, the rhizosphere of *B*. *boehmii* constitutes a source of promising bacteria in terms of plant-beneficial activities such as mobilization and acquisition of nutrients (nitrogen fixation, phosphorus production), mitigation of abiotic stress (bioremediation through siderophores), and modulation of plant hormone levels (and concomitant biocontrol and phytostimulation). Most bacteria belong to a group of newly identified species emerging from recent studies in recalcitrant environments and deserve special attention as they may constitute a new group of PGPB.

## Figures and Tables

**Figure 1 microorganisms-09-01562-f001:**
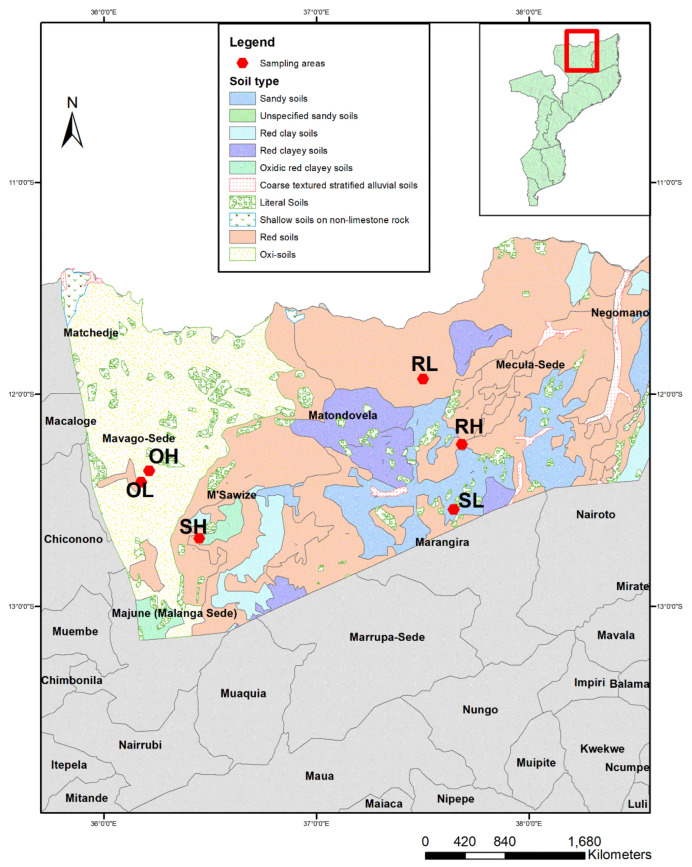
Geographic localization of the sampling sites (red dots) in the Niassa Special Reserve. Sampling points starting with S, R, and O indicate soil types of brownish-gray sandy soils, red soils of medium texture, and red oxi-soils of medium texture, respectively. Sampling points ending with H indicates high fire frequency (fire return interval < 1 year) while L (fire return interval > 7 years) indicates low fire frequency.

**Figure 2 microorganisms-09-01562-f002:**
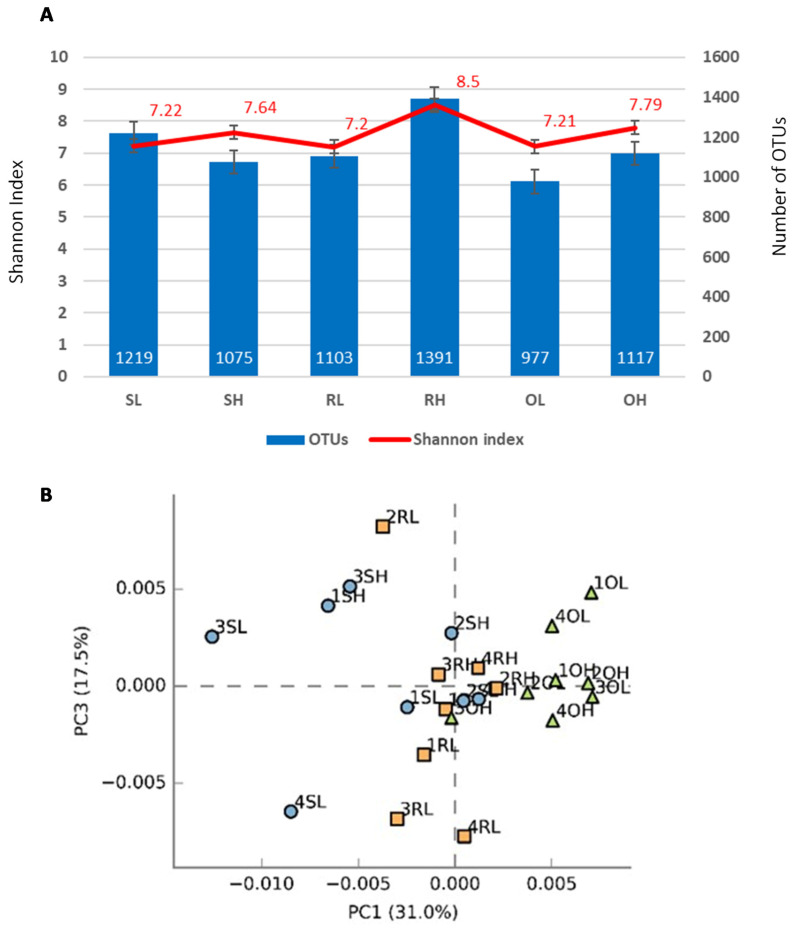
(**A**) Bacterial diversity in soils considering different rhizosphere samples (x-axis). (**B**) Principal component analysis (PCA) of bacterial communities in the different soil samples. S: sandy soils; R: red soils; O: red oxi-soils. H: high fire frequency; L: low fire frequency. Colors preceding the sample codes indicate one type of sample.

**Figure 3 microorganisms-09-01562-f003:**
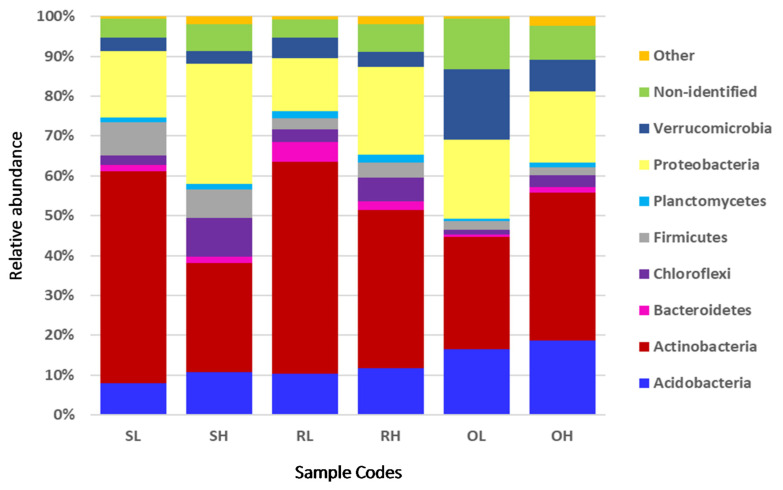
Relative abundance of bacterial phyla in the rhizosphere of *Brachystegia boehmii* sampled in three different soils under low and high fire frequencies. Sample codes follow Figure 1. Rare phyla (<1%) were grouped as Other.

**Figure 4 microorganisms-09-01562-f004:**
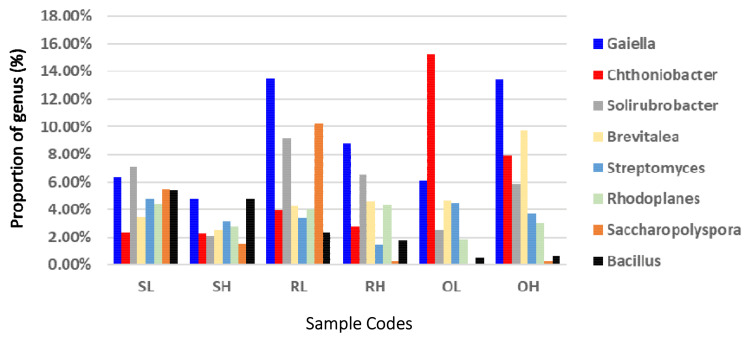
Relative abundance of the most frequent bacterial genera (>2% across all samples) in the rhizosphere of *Brachystegia boehmii.* Sample codes follow Figure 1.

**Figure 5 microorganisms-09-01562-f005:**
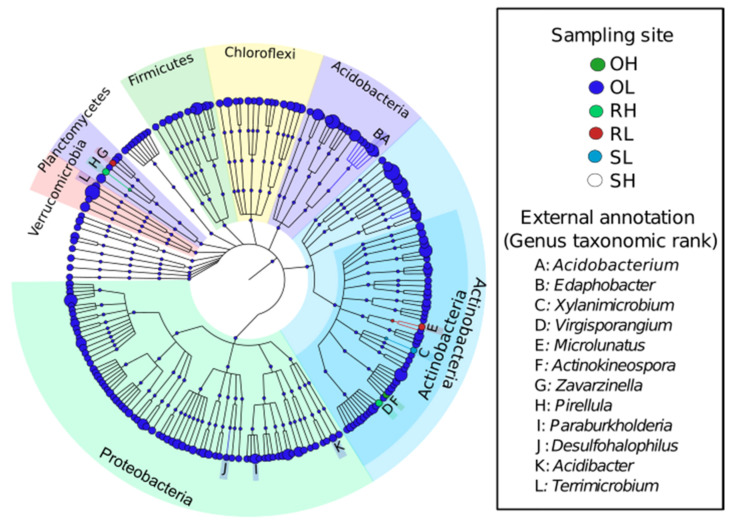
Differentially abundant genera assessed using LEfSE. Only genera with a LDA score > 2.0 and Kruskal-Wallis *p* < 0.05 are displayed.

**Figure 6 microorganisms-09-01562-f006:**
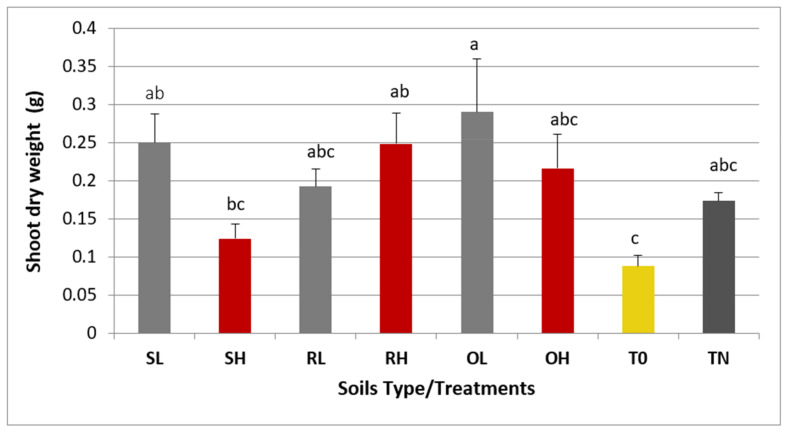
Dry weight of *Vigna unguiculata* plants inoculated with different soil samples. Population codes follow Figure 1. Uninoculated plants supplied either with nitrogen (TN) or without mineral N (T0) were also included. Values of shoot dry weight are the average ±SE of five replicates/soil type. Different letters (a, b, and c) above the columns express significant differences between treatments (including plants inoculated with the soil samples, TN, and T0) according to the Fisher’s LSD test at *p* ≤ 0.05.

**Table 1 microorganisms-09-01562-t001:** Description of soil types and sample codes (underlined Caps).

Soil Type	Fire Frequency	Code
Brownish-gray sandy soils	Low	SL
	High	SH
Red soils of medium texture	Low	RL
	High	RH
Red oxic soils with medium texture	Low	OL
	High	OH

**Table 2 microorganisms-09-01562-t002:** Taxonomical identification of soil isolates from NSR soils and related phylum according to the most related source organism deposited in GenBank. Population codes follow Figure 1. GenBank accession numbers indicate the sequences generated in this study.

Soil Type	Fire Frequency	Isolate	Most Related 16S rRNA Gene Sequence	GenBank Accession Number	% Identity	Growth in N-Free Media	Phosphate Solubilization	Indole Acetic Acid Production ^(1)^	Siderophore Production	Hydrolysis of Cellulose
Sandy soils	Low	10SLA	*Microvirga* sp.	MZ571264	96.54%	+	-	+	-	-
Red soils	High	4RHB	*Caballeronia zhejiangensis* strain ND-B	MZ571257	96.90%	+	-	-	+	-
		5RHB	*Burkholderia* sp.	MZ571258	96.27%	+	-	-	+	-
		6RHB	*Burkholderia* sp. clone P4s-284	MZ571259	72.58%	+	-	+	-	-
Oxi-soils	Low	4OLA	*Rhizobium* sp. NA11036	MZ571262	90.95%	+	-	+	-	-
		5OLA	*Rhizobium altiplani* strain BR 10423	MZ571263	89.23%	+	-	+	-	-
	High	5OHA	*Rhizobium* sp. isolate Moz93	MZ571260	89.70%	+	-	+	-	-
		10OHA	*Variovorax defluvii* strain 2C1-21	MZ571261	97.88%	+	-	-	-	+

^1^ Values greater than 5.0 µg mL^−1^ IAA in the supernatant of cultures grown for 16 h in tryptophan-supplemented TY medium were considered positive.

## Data Availability

The data presented in this study are available in the article and in the Appendix A.

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
