# Peer review of "The Nexus between Fire and Soil Bacterial Diversity in the African Miombo Woodlands of Niassa Special Reserve, Mozambique"

_microorganisms, 2021, doi:10.3390/microorganisms9081562_

Round 1
Reviewer 1 Report
In this manuscript, Maquia et al. studied the fire effects on the soil bacterial diversity in the African miombo woodlands of NSR, Mozambique. They used both Next Generation Sequencing and culture-dependent approaches to analyze the bacterial diversity and associated functions in the rhizosphere of Brachystegia boehmii. They found that bacterial communities in the rhizosphere are highly diverse and driven by soil type and fire regime. Moreover, a diverse pool of diazotrophs was isolated and included symbiotic and non-symbiotic bacteria. Some isolates presented cumulative plant growth-promoting traits
In general, this is a well-designed study. The methods and results are well presented. It can be accepted for publishing after minor modification.
I have several minor comments for the authors as below:
1) It seems to me that the title does not exactly represent the full study. Fire maybe a major factor for the differences in the soil microbiome, however, many other factors may play important roles too. Please modify the tile.
2) Line 39, the conclusion does not show the results from the main study purposes of this study.
3) In the instruction part, please make it clear about the motivation and main purposes.
4) Line 104. It is important to present the soil properties in the main text instead of in the supplemental material.
5) Line 136. where and how the sequencing was done?
6) Line 189. germinated in. what is 0.8% w/v water agar?
7) Line 191. how much in volume soil solution?
8) Line 209. please make it clear how much soil and 0.8% NaCl in volume.
9) Line 224, what is NFb agar?
10) FIgure 2, it is better to show the results of PCA analysis
11) Line 241, what is 3OH?
12) Figure 3. It is good to show the statistical significant difference of these bacterial phyla
13) Figure 4. It is important to indicate the statistical significance.
14) Figure 6 has the same issue. No statistical analysis
15) Line 438, grew in Burk medium
Reviewer 2 Report
The manuscript by Maquia et al. presented studies concerning the characterization of soil bacterial communities in the miombo woodlands. The study is very interesting and the manuscript is well structured and written. However, I have some questions and moreover, it would be advisable to make some corrections before accepting:
- Introduction section– a valuable addition would be to provide short information about the characterization and importance of plant growth-promoting bacteria.
- Section 2.3. DNA extraction (…) – Did The Authors tested the quality and quantity of DNA isolates before library preparation? Please add a short description.
- Figure 2 is very small and cut. Please add the description of the x and y-axis in Figure 2A.
- Figure 3 and Figure 4 – please add a description to the y-axis
- Please provide the assession number to raw sequences deposited in the database (SRA database, GeneBank, or other)
- Section 3 should be named: Results and discussion
Reviewer 3 Report
Revision for manuscript microorganisms-1291111 The nexus between fire and soil bacterial diversity in the African miombo woodlands of Niassa Special Reserve, Mozambique.
Dear authors I have found the manuscript quite interesting, well presented and accurately structured. I only have some minor comments/questions and/or suggestions you can find below:
Lines 105-111: It would be helpful to put this information in a small table in order to visualize the differences between samples or to make a reference to Table S1.
Line 196. Do you mean that samples were incubated at both temperatures indicated or or in cycles from one temperature to another?
Line 202: Why is 28ºC the temperature choosen?
Line 213. I am not sure I have missed something before but you should specify what TY and TYA are.
Line 220. Could you briefly explain why these haloes are indicative or siderophore? (or add a reference for it?)
Line 228. This section is not just “Results” it is “results and discussion” indeed.
Line 226. It is the pyrodiversity effect positive or negative? Please, indicated in the text.
Line 380. Where table A7 is?
Lines 441-442 and 449-450. These are repeated sentences. Also: could you, please, explain in the text why growing in nitrogen-free medium makes bacteria potential nitrogen fixers?
Line 485. Positive or negative effect?
Line 493. Related to which fire frequency it is the diversity that high?
